# GIFF: Generalized Inference Friendly Forward-forward Algorithm

## Abstract

The Forward-Forward (FF) algorithm has recently been proposed to enable neural network training using only forward passes, inspired by the human cortex excitation and inhibition mechanisms. In contrast to Backpropagation (BP), which uses a global loss function, FF utilizes local loss functions at each layer, reducing peak memory requirements. The local weight update scope allows alternative optimizers, non-differentiable neural network layers, and aggressive quantization. Despite its promise for on-device training, the original FF technique has three major limitations that hinder its potential: the label embedding problem, lack of support for convolutional layers, and inefficient inference passes. These issues hamper its performance even on basic datasets like CIFAR10 and restrict its applicability. This paper presents the Generalized Inference Friendly Forward-Forward (GIFF) algorithm to address the limitations of the FF algorithm. We demonstrate GIFF on three representative tinyML benchmarks where FF cannot function. GIFF performs as well as BP on all three tasks, using up to 43% less memory. Furthermore, GIFF requires significantly fewer computations than FF for inference. Thus, GIFF unlocks the potential benefits of the FF algorithm for efficient on-device learning.

## 1 Introduction

Machine learning (ML) applications have become prevalent in many fields, from healthcare to education to entertainment. ML algorithms are traditionally trained on massive datasets on computationally capable server machines (Kang et al., 2017). However, the conventional pipeline of storing data for offline training is prohibitive in edge artificial intelligence (AI) applications that rely on new data collected from sensors (Zhu et al., 2022). According to Semiconductor Research Corporation's decadal plan, digitizing and transmitting the sensor data for offline processing alone will exhaust the world's total energy production by 2030 (SRC, 2020). Hence, there is a strong need to enable continual learning at the edge. Recent *tinyML* research focuses on deploying ML algorithms on resource-constrained edge devices(Warden & Situnayake, 2019; Shafique et al., 2021; Ray, 2022). Most studies on tinyML focus on flows for deployment and efficient inference while limited effort aims to facilitate training on edge, a critical enabler for continual learning (Lin et al., 2022).

State-of-the-art neural network training techniques employ the backpropagation (BP) algorithm that uses stochastic gradient descent. As a result, most tinyML studies aim to enable BP on resource-constrained devices (Patil et al., 2022; Xu et al., 2022; Gim & Ko, 2022; Wang et al., 2022; Profentzas et al., 2023; Ren et al., 2021; Lin et al., 2022). However, BP-based training is fundamentally unsuitable for these devices because BP (i) is memory-hungry, with a significant peak memory requirement; (ii) relies on differentiation, which could be costly for small devices; (iii) has risks of local minima, vanishing, and exploding gradients, which makes quantized approaches challenging. Therefore, a fundamentally different, orthogonal research effort is developing neural network training techniques tailored to on-device learning *by construction*. The most recent such algorithm is the Forward-Forward algorithm proposed by Hinton (2022).

The Forward-Forward algorithm (FF) is inspired by the excitation and inhibition mechanisms in the human cortex. Instead of using a singular global loss function (e.g., cross-entropy, MSE) like BP, FF uses layer-based local loss functions that mimic the excitation in the cortex. Thus, the scope of the gradient updates is much smaller in FF (a single layer), leading to a smaller memory requirement and also making it quantization-friendly. Consequently, FF offers a promising alternative for on-device training. However, the current FF technique has *three major* shortcomings that hinder its usefulness.

**1. Label embedding problem:** FF requires the class labels to be embedded in the input data before feeding them to the model. For example, Figure 1a) and b) illustrate the data and label embedding for MNIST and CIFAR10 datasets. Since the MNIST image is simple and contains many "blank" pixels, the labels are inserted into the first row as a one-hot encoded vector. In contrast, CIFAR10

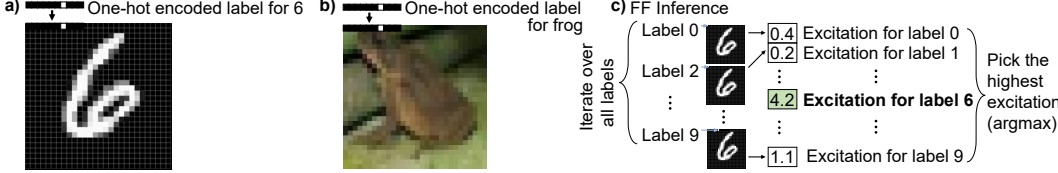

Figure 1: a) Label embedding into an example MNIST image. b) Label embedding into an example CIFAR10 image. c) FF inference iterates over all possible labels in the dataset.

images have a non-trivial background, making this embedding infeasible. One can add a new row or a boundary around the original image, but these ad-hoc choices alter the input images and yield limited success. So, embedding the labels into the data may not be possible for real-world use cases.

**2. Lack of support for convolution layers:** FF cannot reliably train layers with shared weights because of the embedded labels in the data. The shared weights diminish the contribution of the embedded labels in the output activations, which hinders learning and leads to poor accuracy. Thus, FF works with fully connected layers, and crucial layers such as the conv1d and conv2d are left out. FF's performance on standard datasets suffers significantly due to this severe limitation. For example, an FF-based MLP classifier achieves only 60% accuracy on the CIFAR10 dataset (Hinton, 2022).

**3. Inefficient inference passes:** For inference, FF appends all target labels one by one, executes each case, and picks the label that *excites* the network the most, as illustrated in Figure 1c). Each label requires a separate inference since the output activations depend on the label embedded in the data. This requirement leads to highly redundant computations. Moreover, the redundancy scales with the number of target labels in the dataset. For instance, FF requires approximately $10\times$ and $100\times$ FLOPs compared to traditional inference calls on the CIFAR10 and CIFAR100 datasets, respectively.

Addressing these three fundamental limitations is crucial to unlocking the potential benefits of training techniques that employ only forward passes. This paper presents GIFF, the Generalized Inference Friendly Forward-Forward algorithm to address these shortcomings. GIFF builds upon the same principle as FF to maintain its benefits. It employs only forward passes for training. It also uses local (per-layer) custom loss functions for weight updates. In addition, it introduces several structural and algorithmic modifications to address the three limitations described above.

**1. & 2. Independent channel for labels and support for convolution layers:** GIFF features an independent datapath for labels that runs parallel to the input data, eliminating the *label embedding problem* to the input data. This structural change provides more flexibility by decoupling the weights for input data and labels. More importantly, this change enables GIFF to incorporate *weight-sharing layers*, providing support for convolutional layers and standard datasets; hence the term "generalized".

**3. Efficient inference passes:** As mentioned earlier, the FF algorithm performs a separate inference pass for each label in the dataset. GIFF overcomes this limitation since the output activations are calculated using the *separated* input data and label activations. As a result, the activations for the input data can be calculated just once, similar to BP. Then, activations only for the label channels can be computed incrementally. This feature eliminates the redundant computations necessary for FF's inference, hence the term "inference friendly".

In summary, GIFF addresses the three significant limitations of FF, generalizing it to arbitrary neural networks (e.g., CNN) and making it inference-friendly. Appendix A.1 theoretically shows that GIFF indeed presents a generalization of the original FF algorithm. Given a trained FF network, there exists a set of weights that guarantees the output activations of GIFF are equal to that of FF. Our experimental evaluations show that training the GIFF network directly gives the same or superior results and support the following observations: **First**, GIFF is superior to FF in cases where they both work (e.g., MLPs). For example, Section 3.4 shows that GIFF achieves identical accuracy as FF while executing $10\times$ smaller FLOPs for inference. More importantly, GIFF works on a broader range of problems that FF cannot handle. For instance, we show that GIFF can train ResNet and MobileNet architectures commonly used in tinyML tasks. **Second**, GIFF enables side-by-side comparisons with state-of-the-art backpropagation (BP)-based training, which has been exhaustively studied and optimized. Section 4 shows that GIFF achieves competitive accuracy as BP while requiring up to 43% smaller peak memory requirement during training on three representative tinyML benchmarks, offering a promising alternative to BP for on-device training. *Thus, our major contributions are:*

- GIFF, a novel training algorithm that addresses the major limitations of FF, generalizing it to arbitrary neural networks and making it inference friendly,
- Detailed quantitative exploration and comparison of GIFF with FF and BP on three representative tinyML benchmark datasets,
- Open-source implementation of the FF and GIFF algorithms in Python and C.

## 2 RELATED WORK

TinyML frameworks cater to the unique requirements of deploying machine learning models on resource-constrained microcontrollers. Commonly used frameworks include TensorFlow Lite for Microcontrollers (TFLM) (David et al., 2021), μTensor (2018), Glow (Rotem et al., 2019), X-Cube-AI (STMicroelectronics, 2018), Plumerai (2017), and μTVM (Reusch, 2021). TFLM is a lightweight version of TensorFlow tailored for inference on microcontrollers and tiny devices. μTensor is an embedded machine learning inference library for rapid prototyping and deployment. Glow is a compiler framework that accelerates the performance of neural network frameworks on different hardware platforms, including microcontrollers. X-Cube-AI enables the automatic conversion and optimized deployment of neural networks and other classical ML models on STM32 microcontrollers. Similarly, the Plumerai inference engine is a compiler designed to optimize and accelerate deep learning model execution on resource-constrained microcontrollers. Finally, μTVM is a specialized version of the TVM compiler stack, tailored for efficiently deploying models on microcontrollers and other resource-limited devices. These frameworks prioritize reduced model sizes and improved inference performance. They also provide streamlined flows to simplify the deployment of the learned model. However, they *do not* facilitate training at the edge.

Recent research has also addressed efficient on-device training at the edge. These studies include POET (Patil et al., 2022), Mandheling (Xu et al., 2022), SAGE (Gim & Ko, 2022), Melon (Wang et al., 2022). POET finds the most energy optimal schedule to train *large models* on microcontrollers by rematerializing or paging the tensors to secondary storage. Mandheling enables resource-efficient on-device training by judiciously co-scheduling various training operators to mobile DSP and CPU. SAGE is a framework for efficiently optimizing memory resources for on-device DNN training. Melon is an on-device learning framework that enables training with large batch sizes beyond the physical memory capacity. Although relevant, the above studies focus on training large models on devices such as Nvidia Jetson, Raspberry Pi, and Smartphones which typically have stronger cores and more memory than microcontrollers. Other recent research focuses on training on microcontrollers, such as MiniLearn (Profentzas et al., 2023), TinyOL (Ren et al., 2021), and TTE (Lin et al., 2022). MiniLearn is a system to re-train or optimize pre-trained, quantized DNNs on resource-constrained IoT devices. TinyOL enables incremental on-device training using new streaming data. TTE is a collection of improvements to enable on-device training with minimal memory footprint. All of these studies aim to reduce the execution time and memory footprint of the backpropagation algorithm. In contrast, we propose GIFF, a *fundamental alternative to the backpropagation algorithm*.

After the Forward-Forward algorithm appeared, several follow-up techniques have been proposed in less than a year. SymBa (Lee & Song, 2023) aims to improve the FF algorithm's convergence rate by modifying the label embedding technique and the loss functions. Similarly, PFF (Ororbia & Mali, 2023) dynamically generates new negative data samples to boost FF's learning performance. Another recent technique, FFCL (Ahamed et al., 2023), shows that using FF as a pretraining strategy for backpropagation-based training boosts the final accuracy of the model. Most other studies on FF focus on exploring novel loss and goodness functions, generating negative data, and utilizing FF as novel feature extractors. GIFF is the first study to address the primary limitations of FF: Lack of support for convolutional layers and the inefficient inference pass.

## 3 OVERVIEW

### 3.1 THE BACKPROPAGATION ALGORITHM

The BP algorithm is the *de facto* technique for training deep neural networks in numerous domains, from image classification to generative tools like ChatGPT. Its remarkable ability to learn the network weights based on errors between predictions and desired outcomes established BP as the fundamental tool in deep learning. The learning during the training phase occurs in two passes: The forward pass and the backward pass, as shown in Figure 2a). The forward pass propagates the input data through the network and produces output and intermediate activations, which are stored for the backward pass. A loss function quantifies the error between the network prediction produced by these activations and the target labels. The backward pass minimizes this error by backpropagating it and updating *all*

network parameters using the stored activations and gradients. Despite its enormous and successful track record, the backpropagation approach has several drawbacks, which have been discussed by many researchers (Hinton, 2022; Eshraghian et al., 2023; Lin et al., 2022), as summarized in Table 1:

- BP requires all activations computed during the forward pass to be stored to calculate the derivatives, making it memory-hungry,
- BP requires differentiable functions since it uses gradient descent for weight updates, making it susceptible to problems of local minima, vanishing and exploding gradients.

These drawbacks and concerns motivated researchers to explore more efficient and biologically plausible learning paradigms to approach the efficiency of the human cortex. Moreover, BP training and its inference are strictly sequential, i.e., hard to pipeline or parallelize. The most recent and promising alternative paradigm is the Forward-Forward algorithm (Hinton, 2022).

### 3.2 THE FORWARD-FORWARD ALGORITHM

Forward-Forward algorithm is a recent novel technique inspired by the neural activity in the brain. As illustrated in Figure 2b), FF embeds the labels and data into a single representation before feeding it to the network instead of forward-propagating only the data. In contrast to the forward and backward passes in backpropagation, FF employs two forward passes that operate in exactly the same way as each other but on different data and with opposite objectives: first using *positive data*, and the second using *negative data*. The *positive data* refers to when data and the label are taken from the training data distribution, whereas *negative data* uses corrupted data, incorrect labels, or both. In each pass, FF updates the local parameters in every network layer using individual *goodness* and *loss* functions. Alternatively, the activations from the two passes can be aggregated, and a combined loss function for positive and negative data can be used to update the weights (Lee & Song, 2023). The goodness and loss functions should be designed such that positive data "excites" the neurons (i.e., high goodness) and negative data "inhibits" the neurons (i.e., low goodness). For instance, the goodness function can be simply the sum of squared activations, and the loss can be the logistic function:

$$g = \sum_{j=1}^{J} h_j^2 \quad (1) \qquad \begin{aligned} Loss(Pos) &= log(1 + exp(\theta - g)) \\ Loss(Neg) &= log(1 + exp(g - \theta)) \end{aligned} \quad (2)$$

where $h$ is the vector/matrix of activations with $J$ elements, $\theta$ is the "threshold" selected according to expected value of $g$ in that layer, and $Pos$ and $Neg$ denote the positive and negative data. Minimizing the above losses corresponds to maximizing the goodness for positive data and minimizing the goodness for negative data. As a result, each layer in the network gets "excited" with positive data and "inhibited" with negative data, thereby learning to differentiate positive from negative data. During inference, *FF iterates over all labels* for each sample and choose the label that yields the maximum collaborative goodness across all layers. FF addresses most of the concerns pertaining to BP. Namely,

- Activations *from only one layer* are stored for a batch of inputs since FF uses local updates, significantly reducing its memory requirements,
- FF does not depend on gradient descent for weight updates. In contrast, it can use any optimization technique, enabling non-differentiable layers (e.g., decision trees), alleviating local minima risk and vanishing/exploding gradient issues.

Hence, FF layers can be parallelized since it can process the next batch of inputs as soon as the previous batch completes. However, the original FF technique still has several critical limitations:

- The embedding of labels and data is arbitrary and can be tricky (e.g., non-trivial backgrounds),
- Weight-sharing layers are not supported, preventing training of widely used networks (e.g., CNNs),
- Inference is highly inefficient due to repeated iterations for all possible labels (e.g., an $N$-class classification problem requires $N$ inferences per sample for a prediction).

Table 1: Summary of advantages and disadvantages of BP, FF and GIFF.

|  | **BP** | **FF** | **GIFF** |
|---|---|---|---|
| **Scope of Training** | Global | Local (layer-wise) | Local (layer-wise) |
| **Optimizer** | Differentiable | Arbitrary | Arbitrary |
| **Requires label embedding** | No | Yes | No |
| **Support for Conv. Layer** | Yes | No | Yes |
| **Inference efficiency** | High | Low | High |

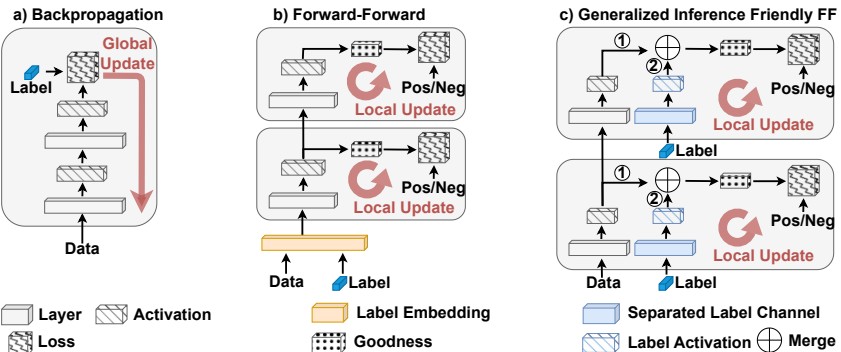

Figure 2: Overview of the a) Backpropagation algorithm, b) Forward-Forward algorithm, and c) GIFF. ①: Data activations. ②: High-dimensional latent label representation.

## 3.3 GIFF: GENERALIZED INFERENCE FRIENDLY FORWARD-FORWARD ALGORITHM

GIFF maintains the positive and negative data distributions and the goodness function introduced by the FF algorithm. However, the data and the label are not embedded in the beginning, as illustrated in Figure 2c). Instead, GIFF feeds data and labels as two separate channels and uses the merging operator $\bigoplus$ after the nonlinear activation function in each layer to merge the latent data and label representations. The merged latent is then used to compute the goodness, and the loss function is designed to maximize (minimize) the goodness for positive (negative) data, similar to FF. There are four main components in GIFF; (i) the input data channel, (ii) the label channel, (iii) the latent merging operator $\bigoplus$, and (iv) the goodness and loss functions.

**The input data channel:** In contrast to FF, GIFF separates the input and label data channels. The input data and label data pass through dedicated computational layers and activation functions. The data activations can be normalized before propagating to subsequent layers, as suggested in the original FF (Hinton, 2022). As a result of separated channels, GIFF does not have limitations on the types of layers that can be used. Consequently, a network architecture implemented for BP can be used for the data channel, as depicted in Figure 2a) and c). For example, *GIFF can train ResNet and MobileNet models, which is not possible with FF*.

**The label channel:** Since the class labels are passed through a dedicated channel parallel to the input data, the labels are transformed to a higher dimension controlled by the parameters of the computational layer. This high-dimensional latent label representation and the activations in the data channel are then passed to the merging operator $\bigoplus$. GIFF is oblivious to the format of the label; it can be decimal, tokenized, or one-hot encoded. The label is *broadcasted* to all layers, which is different from the data channel, where activations are propagated to the next layer. This implementation choice reduces the number of parameters in the label channels and hence, in the overall network, since the dimension of the label is smaller than the high-dimensional latent labels.

**The latent merging operator $\bigoplus$:** In a layer, the activations in the data channel and the latent labels are combined through the merging operator. The choice of operator $\bigoplus$ is *task-specific*. For example, simple addition ($+$) for MLPs yields sufficient performance. However, elementwise multiplication ($\times$) yields higher performance for image-based networks that rely on 2D convolutional layers. In this case, the latent labels serve as a mask that highlights the important features in the data. The merging can also be performed with a custom user-defined operator.

**The goodness and loss functions:** GIFF does not constrain the goodness and loss functions choices. The positive and negative data should push the goodness in opposite directions so that the network can correctly classify input vectors as positive or negative. In our experiments, we use Equations 1 and 2 for goodness and loss functions, respectively. We emphasize that these functions can be customized to obtain higher accuracy for a given dataset. We do not explore this customization since this aspect does not differ from the original FF algorithm. This work aims to introduce GIFF, highlight its advantages compared to FF, and provide example side-by-side comparisons to BP.

## 3.4 MOTIVATIONAL EXAMPLE: MNIST DIGIT CLASSIFICATION

We use the MNIST dataset of handwritten digits as a motivational example to demonstrate the benefits of using GIFF compared to FF. Figure 3 illustrates the differences in the same architecture between

the FF and GIFF algorithms. We also include the BP algorithm in our results to serve as a reference and side-by-side comparison. MNIST has been well studied, and the performance of simple neural networks trained with BP is well known as a reference point. We employ the official 50,000 training and 10,000 test images in MNIST without introducing any data augmentations.

**Comparisons on an MLP:** Our quantitative comparisons start with a 3-layer MLP architecture with 1000 neurons in each layer since it is used in the original FF study (Hinton, 2022). Table 2a) shows that GIFF, FF, and BP all achieve 97%+ accuracy with about 2.8M parameters. The notable difference is the 32% lower peak memory usage of FF and GIFF (15.3 MB) compared to BP (22.3 MB).

**Comparisons on a CNN:** CNNs can achieve similar levels of accuracy as MLPs using drastically fewer parameters due to shared weights. *However, FF does not work with any layers that use weight-sharing*, as mentioned before. In strong contrast, GIFF supports any type of layers. Indeed, Table 2b) shows that GIFF achieves similar accuracy as BP while consuming less memory. More importantly, Table 2 shows that GIFF enables higher accuracy than FF (98.4% vs. 97.2%) using 279× fewer parameters (2.82M vs. 10.1k) and 220× smaller peak memory (15.3 MB vs. 73.4 kB).

Table 2: Comparison of GIFF, FF, and BP for MNIST dataset on **a) 3-layer MLP**, **b) 3-layer CNN.**

| a) | GIFF | FF | BP | | b) | GIFF | FF | BP |
|---|---|---|---|---|---|---|---|---|
| # Of Params | 2.79M | 2.82M | 2.78M | | # Of Params | 10.1k | N/A | 10.9k |
| Peak Memory | 15.3 MB | 15.3 MB | 22.3 MB | **GIFF can train CNNs** → | Peak Memory | 73.4 kB | N/A | 108 kB |
| Accuracy | 98.0% | 97.2% | 97.9% | | Accuracy | 98.4% | N/A | 98.8% |

Another advantage of GIFF over FF is the significantly reduced inference cost. For a given test data sample, FF iterates over all the layers in the dataset, as shown in lines 2–6 of Algorithm A1. In each iteration, it embeds the label in the data (line 3), passes the embedded data through the network to obtain activations (line 4), calculates the goodness per layer using the activations (line 5), and stores the total goodness in an array (line 6). Then, it chooses the label that maximizes the overall goodness of the network (i.e., excitation) as the predicted label (line 7). This requirement leads to many redundant computations compared to the traditional inference using the softmax outputs.

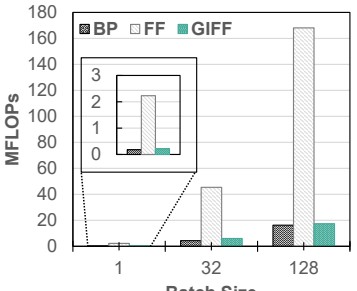

Figure 4: MFLOPs executed for inference in GIFF, FF, and BP.

GIFF reduces the inference cost of the FF algorithm by pre-computing and storing the activations for the separated label channel, as shown in Algorithm A2. Consequently, *GIFF calculates the data activations just once* (line 6). Then, it iterates over the labels (lines 7–10), merges data activations with the corresponding stored label activation using the merging operator $\bigoplus$ (line 8), calculates the goodness for every layer (line 9), and keeps the total goodness in an array(line 10). As a result, total computations are similar to the traditional inference, as demonstrated in Figure 4. GIFF has about 10× fewer MFLOPs than FF and is almost identical to BP.

In summary, GIFF generalizes the FF algorithm to CNNs and other weight-sharing networks and reduces the computations required for inference to the same level as BP. Hence, GIFF addresses

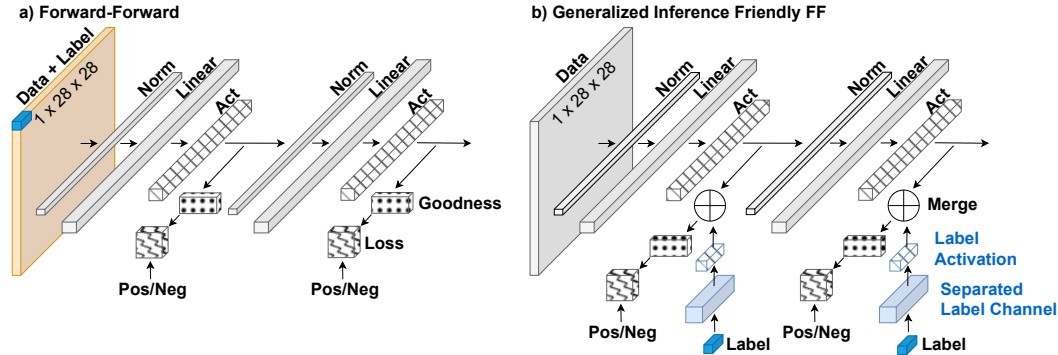

Figure 3: The MLP for a) FF and b) GIFF for MNIST dataset (third layer is omitted for space).

FF's major limitations and unlocks its potential as a viable alternative for BP-based training on edge devices. The rest of this paper shows that GIFF is comparable or superior to BP, even with standard gradient descent algorithms. We stress that these algorithms and optimizers have been tailored for efficient backpropagation, which has been around for a few decades. In contrast, GIFF can support different kinds of optimizers, such as evolutionary-based or one-shot optimizers, which can further facilitate the efficiency of GIFF for on-device training.

## 4 EXPERIMENTAL EVALUATION

This section demonstrates the applicability of GIFF on three representative tinyML benchmarks. We compare networks trained by GIFF and BP for each benchmark regarding accuracy, number of floating point operations, and memory usage. We exclude the accuracy of FF models because FF does not support the models used in this section due to its previously mentioned limitations.

### 4.1 EXPERIMENTAL SETUP

MLPerf Tiny benchmarks are the state-of-the-art representative set of deep neural networks and benchmarking code tailored for embedded devices (Banbury et al., 2021). We use image classification, person detection, and keyword spotting classification tasks as our target benchmarks, as summarized in Table 3. Each task has a corresponding dataset, model, and target accuracy value.

Table 3: Benchmark tasks and corresponding datasets, models, and target accuracy metrics.

| Task | Dataset | Model | Target Accuracy | GIFF |
|---|---|---|---|---|
| Image classification | CIFAR10 | ResNet-8 | 85% | 85.6% |
| Person detection | Visual Wake Words (VWW) | MobileNetV1 | 80% | 81.3% |
| Keyword spotting | Google Speech Commands (GSC) | DS-CNN* | 90% | 90.4% |

*DS-CNN: Depthwise Separable Convolutional Neural Network

For each benchmark, we implement the baseline BP training using the networks provided by MLPerfTiny benchmarks[1] (listed in Table 3).The proposed GIFF technique also uses the same baseline model for its data channel, as illustrated in Figure 2c). Then, we sweep the GIFF-specific hyperparameters: threshold $\theta$ in the loss function (Equation 2) and the label channel width, as detailed in Appendix A.3. By default, GIFF trains each layer individually, i.e., each layer has distinct label channel, goodness and loss function, and optimizer. GIFF also enables training layers in groups to reduce parameter count and memory requirements. For example, the first three consecutive layers can be optimized at once by evaluating the goodness and loss functions at the output of the third layer. Hence, we also explored different layer groups, as summarized in Table A1 in Appendix A.3.

After parameter sweeps, we identify the top 100 configurations with the highest test accuracy and record their total number of parameters, estimated peak memory consumption during training, test accuracy, and the number of single precision floating point operations (FLOPs) for inference.

**Total number of parameters:** We use the open source `torchinfo`(Yep, 2020) library to obtain a detailed breakdown of the total number of trainable parameters in a pytorch model.

**Peak memory consumption:** We estimate the memory requirement for training the BP and GIFF models. Appendix A.4 presents our estimation method and shows that our estimations are within 5% of a C implementation where memory allocations are manually implemented.

**Test accuracy:** We report the accuracy on the test sets provided by MLPerf Tiny for all three tasks.

**FLOPs:** We record the performance counter information on our system using the Performance Application Programming Interface (PAPI) version 7.0.1 (PAPI, 2023). FLOPs information is retrieved using the "PAPI_SP_OPS" counters during the function calls to `inference_FF()` and `inference_GIFF()` given in Algorithms A1 and A2, respectively. The experiments are run on an isolated machine to minimize interference by other workloads. The recorded values are averaged over the entire test set, as some of the computations may be optimized out for some inputs.

All models are implemented in Python 3.9 using Pytorch v1.12.1. They use single-precision floating point numbers without quantization during and after training. We use SGD with Adam optimizer to update weights with a learning rate of 0.001. GIFF layers can be trained with other optimization techniques, such as evolutionary optimizers and integer linear programming (ILPs). Our evaluations use SGD for an apple-to-apple comparison with BP. Exploring different optimizers and the impact of

---

[1] https://github.com/mlcommons/tiny

quantization on GIFF networks is left for future work since this paper aims at introducing GIFF and comparing it to FF and BP-based learning. Finally, the training experiments are assigned to a data center that houses different types of GPUs, such as A100, P100, 2080Ti, and L40.

## 4.2 IMAGE CLASSIFICATION: CIFAR10 DATASET WITH THE RESNET-8 ARCHITECTURE

MLPerf Tiny uses the ResNet-8 architecture for the CIFAR10 image classification benchmark with a target accuracy of 85%. ResNet (Residual Network) architectures contain "residual connections" that allow the output of one layer to bypass intermediate layers. These connections mitigate the vanishing gradient problem, enabling the training of deeper networks. While the original ResNet models can have dozens of layers, ResNet-8 has three residual blocks, one containing two convolutional layers and the other two containing three convolutional layers, making it eight convolutional layers in the blocks, hence the name. It also has an initial convolutional layer to make it a total of 9 layers. This baseline ResNet-8 model has 78k parameters, requires 1227 KB of memory for BP training, and achieves 85% test accuracy, as shown in Figure 5 with the diamond symbol.

GIFF trains the same baseline ResNet network for its data channel. In addition, GIFF has a separate label channel, as described in Section 3.3. To characterize the potential of GIFF over a range of parameters, we sweep the threshold parameter $\theta$, label channel width, and layer groups, as described in Section 4.1. We place the initial convolutional layer and the first residual block in the same group, leading to three groups (Group 1: The first convolutional layer and residual block; Group 2: the second residual block; Group 3: the third residual block). As a result, there are three groups of layers, as summarized in Table A1 and illustrated in Appendix A.3. We will also list all sweep iterations in a spreadsheet in the project repository.

Figure 5 shows the accuracy of different GIFF model configurations and their convex hull. The top left corner in this plot is the ideal solution (highest accuracy and lowest memory usage). GIFF covers a wide range of solutions, including a configuration ① with slightly higher accuracy than BP and only 2% higher peak memory usage. Most notably, it can achieve ② the target accuracy of 85% with 12% less memory usage than BP. It can also use ③ 986 kB memory (20% lower than BP) and achieve 84% test accuracy. Finally, it can use ④ as low as 680 kB memory (45% lower than BP) and still achieve 75% test accuracy.

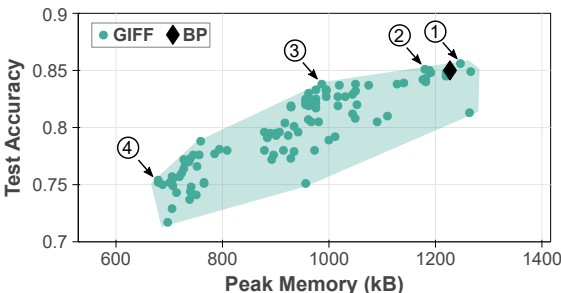

Figure 5: Accuracy vs. peak memory usage of GIFF networks on the CIFAR10 dataset with ResNet-8 network. The BP point is obtained using the parameters provided by the tinyML benchmark.

Inference tasks in this work use three different batch sizes: 1, 32, 128. A batch size of 1 is the most representative tinyML use case, where an inference call happens on a single data point. Larger batch sizes are included to demonstrate that GIFF's benefits extend beyond tinyML. We also measure the FLOPs of the original FF technique's (Hinton, 2022) inference call as an additional data point to GIFF and BP. *Since the FF algorithm cannot train this network architecture, the weights are initialized randomly*. Table 4a) shows that GIFF has a similar FLOP count to BP during inference. This result is as expected since GIFF has marginal extra operations for executing the label channel, merging the activations, and calculating the goodness. In strong contrast, GIFF requires about 10× fewer FLOPs than the FF algorithm for all three batch sizes, proving it successfully addresses the inefficient inference pass problem of the FF algorithm.

Table 4: MFLOPs during inference for a) ResNet-8, b) MobileNetV1, and c) DS-CNN. Since FF cannot train these networks, we measured its FLOPs using random weights.

| | a) ResNet-8 | | | GIFF vs. FF Speedup | b) MobileNetV1 | | | GIFF vs. FF Speedup | c) DS-CNN | | | GIFF vs. FF Speedup |
|---|---|---|---|---|---|---|---|---|---|---|---|---|
| | BP | FF | GIFF | | BP | FF | GIFF | | BP | FF | GIFF | |
| **1** | 2.4 | 31.7 | 3.1 | 10.2× | 1.5 | 6.9 | 3.2 | 2.2× | 0.4 | 5.3 | 0.4 | 12.2× |
| **32** | 27.0 | 289.1 | 29.2 | 9.9× | 22.4 | 51.5 | 25.5 | 2.0× | 6.4 | 75.4 | 6.5 | 11.6× |
| **128** | 105.9 | 1085.6 | 109.5 | 9.9× | 81.3 | 176.1 | 87.6 | 2.0× | 24.4 | 292.5 | 25.0 | 11.7× |

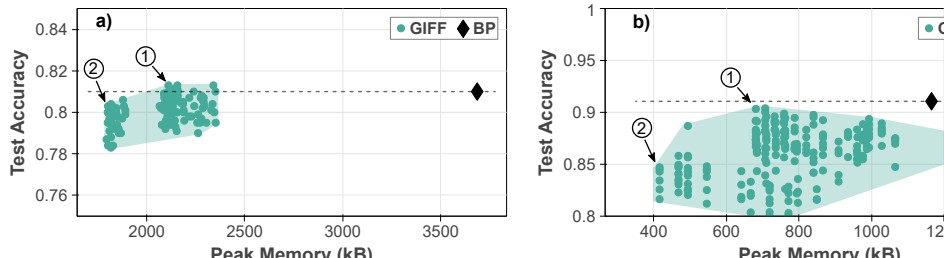

Figure 6: Accuracy vs. peak memory usage of GIFF networks on a) VWW dataset with MobileNetV1 b) GSC dataset with DS-CNN.

### 4.3 PERSON DETECTION: VWW DATASET WITH THE MOBILENETV1 ARCHITECTURE

MLPerf Tiny uses the MobileNetV1 architecture for the VWW person detection benchmark with 80% target accuracy. This is a one-vs-all binary image classification task, where the aim is to detect the presence of a human in a given image. MobileNets are based on depth-wise separable convolutions and typically have more layers than ResNets, which enlarges the design space for parameters specific to GIFF. The baseline MobileNetV1 model has 211k parameters, requires 3686 kB of memory for BP training, and achieves 81% test accuracy, as shown in Figure 6a) with the diamond symbol.

Like ResNet, we sweep the GIFF-specific parameters listed in Table A1 for MobilenetV1. Since MobileNetV1 has a maximum of 13 groups (see Figure A4), GIFF has a larger design space, leading to significant memory savings. Figure 6a) shows GIFF can achieve ① slightly higher accuracy than BP with about 43% smaller peak memory. Moreover, GIFF can achieve ② the 80% target accuracy with only 1800 kB of memory (52% savings compared to BP). Finally, Table 4b) shows GIFF and BP again have similar FLOPs during inference calls for the MobileNetV1 network. In this case, GIFF requires about $2\times$ fewer FLOPs for all three batch sizes than FF, proving it is inference-friendly.

### 4.4 KEYWORD SPOTTING: GSC DATASET WITH DS-CSNN ARCHITECTURE

MLPerf Tiny uses a DS-CNN network for the GSC keyword spotting benchmark with a target accuracy of 90%. GSC is an audio dataset built for training and evaluating keyword spotting systems, and the downsized version for the tinyML benchmark contains Mel spectrogram images for 12 classes. The DS-CNN network uses depth-wise separable convolutional layers, similar to the MobileNet network. The baseline model has 24k parameters, requires 1165 kB of memory for BP training, and achieves 91% test accuracy, as shown in Figure 6b) with the diamond symbol.

Like the previous two cases, we sweep the GIFF-specific parameters for the DSCNN network, as listed in Table A1. The DS-CNN network has a maximum of four groups, as illustrated in Figure A5. Figure 6b) shows GIFF can achieve ① 90.4% accuracy (almost identical to BP) while requiring about 40% less memory. Moreover, GIFF can achieve ② 85% accuracy with only 416 kB of memory, which is about 64% less than BP's memory requirement. Finally, Table 4c) shows GIFF and BP again have similar FLOPs for their inference calls for the DS-CNN network. In this case, GIFF requires about $12\times$ fewer FLOPs for all three batch sizes than FF, proving it is inference-friendly.

## 5 CONCLUSION AND FUTURE WORK

This paper presented GIFF, the Generalized Inference Friendly Forward-Forward algorithm, to address the major limitations of the recent Forward-Forward algorithm (Hinton, 2022). GIFF does not require label embedding like FF, enabling convolutional layers and other weight-sharing layers. For example, GIFF is able to train ubiquitous neural network architectures like ResNet and MobileNet and achieve identical accuracy as backpropagation-based training. Moreover, GIFF is inference-friendly since it does not iterate over the class labels in the dataset like FF. As a result, GIFF executes significantly fewer FLOPs for inference than FF, almost identical to what a traditional softmax inference call requires. Thus, GIFF unlocks the potential of the FF algorithm for efficient on-device learning.

The potential benefits of FF, hence GIFF, include the support for alternative optimizers that do not rely on gradients. Consequently, the harmful impact of quantization on gradient based weight updates is also eliminated. As a result, GIFF also facilitates the training of non-differentiable layers (e.g., decision trees) as part of the neural network. For example, an integer linear programming based optimizer can directly calculate integer weights that minimize the layer's loss. Since the main aim of the current work is to introduce the GIFF algorithm and to highlight its advantages compared to FF, we have left the exploration of the above topics as future work.

ETHICS STATEMENT

This work introduces a novel on-device learning approach based on the recently published Forward-Forward algorithm. Continual on-device learning facilitates user privacy by eliminating the need for transmitting user data to the cloud. Therefore, our work can help alleviate ethical concerns related to sharing sensitive user data.

REPRODUCIBILITY STATEMENT

Proofs of our theoretical results are given in Appendix A.1. Appendix A.2 presents the pseudocodes for FF and GIFF's inference function calls. Appendix A.3 summarizes GIFF-specific parameters and the parameter sweep configurations for our experimental results. Appendix A.4 presents our memory requirement estimation technique. Finally, Appendix A.5 and A.6 illustrate the layer groups for the ResNet-8, MobileNetV1 and DS-CNN networks. The source code is attached with the rest of the supplementary material, providing end-to-end working scripts for the networks and datasets covered in our work.

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

# A APPENDIX

## A.1 CALCULATING GIFF WEIGHTS FROM A TRAINED FF LAYER

A typical FF layer with ReLU as the activation function is shown in Figure A1a). Here, $\mathcal{D} \in \mathbb{R}^{1 \times D}$ depicts the input data vector, and $\mathcal{C} \in \mathbb{R}^{1 \times C}$ denotes the class label vector (e.g., one-hot encoded label). In FF, $\mathcal{D}$ and $\mathcal{C}$ are embedded together (e.g., added, concatenated). Here, we assume $\mathcal{D}$ and $\mathcal{C}$ are concatenated together for embedding and denote the embedded result as $[\mathcal{D}\mathcal{C}] \in \mathbb{R}^{1 \times (D+C)}$.

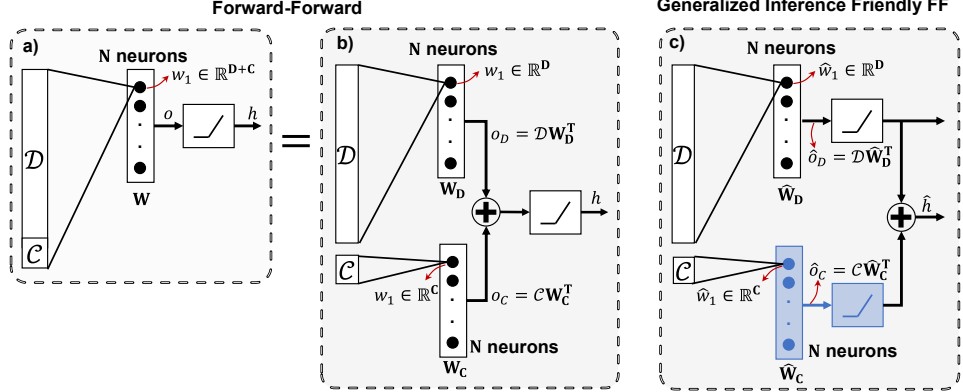

Figure A1: a) An original Forward-Forward input layer, b) Equivalent illustration for easier comparison to GIFF, and c) a typical GIFF layer.

**Obtaining activations of the FF layer:** The linear layer in this case has $N$ neurons, and each neuron has $D + C$ weights. We denote the weight matrix with $\mathbf{W} \in \mathbb{R}^{N \times (D+C)}$. The output of the linear layer is:

$$o = [\mathcal{D}\mathcal{C}]\mathbf{W^T} \text{ where } o \in \mathbb{R}^{1 \times N} \tag{3}$$

If we separate the weights of the neurons that multiply $\mathcal{D}$ and the $\mathcal{C}$, as shown in Figure A1b), the same $o$ can be obtained by:

$$
\begin{aligned}
o_D &= \mathcal{D}\mathbf{W_D^T} && \text{where } \mathbf{W_D} \in \mathbb{R}^{N \times D} \text{ and } o_D \in \mathbb{R}^{1 \times N} \\
o_C &= \mathcal{C}\mathbf{W_C^T} && \text{where } \mathbf{W_C} \in \mathbb{R}^{N \times C} \text{ and } o_C \in \mathbb{R}^{1 \times N} \\
o &= o_D + o_C
\end{aligned}
\tag{4}
$$

We use this notation from hereon, as it makes comparison with GIFF structure easier. The activations $h \in \mathbb{R}^{1 \times N}$ at the output of the FF layer is therefore given by:

$$
\begin{aligned}
h = f(o) &= f(o_D + o_C) \\
&= f(\mathcal{D}\mathbf{W_D^T} + \mathcal{C}\mathbf{W_C^T})
\end{aligned}
\tag{5}
$$

where $f$ denotes the ReLU activation function.

**Obtaining activations of the GIFF layer:** We illustrate the GIFF layer in Figure A1c). Similar to the FF layer, we denote the outputs of the data channel and the separated label channel as follows:

$$
\begin{aligned}
\hat{o}_D &= \mathcal{D}\hat{\mathbf{W}}_{\mathbf{D}}^{\mathbf{T}} && \text{where } \hat{\mathbf{W}}_{\mathbf{D}} \in \mathbb{R}^{N \times D} \text{ and } \hat{o}_D \in \mathbb{R}^{1 \times N} \\
\hat{o}_C &= \mathcal{C}\hat{\mathbf{W}}_{\mathbf{C}}^{\mathbf{T}} && \text{where } \hat{\mathbf{W}}_{\mathbf{C}} \in \mathbb{R}^{N \times C} \text{ and } \hat{o}_C \in \mathbb{R}^{1 \times N}
\end{aligned}
\tag{6}
$$

Finally, the activations $\hat{h} \in \mathbb{R}^{1 \times N}$ at the output of the GIFF layer is given by:

$$
\begin{aligned}
\hat{h} &= f(\hat{o}_D) + f(\hat{o}_C) \\
&= f(\mathcal{D}\hat{\mathbf{W}}_{\mathbf{D}}^{\mathbf{T}}) + f(\mathcal{C}\hat{\mathbf{W}}_{\mathbf{C}}^{\mathbf{T}})
\end{aligned}
\tag{7}
$$

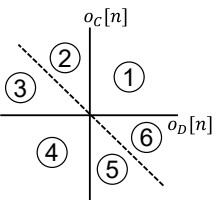

**Can we find $\hat{\mathbf{W}}_{\mathbf{D}}^{\mathbf{T}}$ and $\hat{\mathbf{W}}_{\mathbf{C}}^{\mathbf{T}}$ such that $h = \hat{h}$?** This question is not trivial to answer as $f()$ is the nonlinear ReLU function, which implies $f(a + b)$ is not necessarily equal to $f(a) + f(b)$. Therefore, we need to iterate over the activations for each neuron $n \in [1, 2, ..., N]$ and analyze the problem in the six regions illustrated in Figure A2 for the domain of $f(o_D[n] + o_C[n])$ (i.e., Equation 5):

Figure A2: 6 regions of interest for the domain of $f(o_D[n] + o_C[n])$

① $o_D[n] \geq 0$ and $o_C[n] \geq 0$

$h[n] = f(o_D[n] + o_C[n]) = o_D[n] + o_C[n] = \mathcal{D}\mathbf{W}_{\mathbf{D}}^{\mathbf{T}}[\mathbf{n}] + \mathcal{C}\mathbf{W}_{\mathbf{C}}^{\mathbf{T}}[\mathbf{n}]$

$\hat{h}[n] = f(\hat{o}_D[n]) + f(\hat{o}_C[n]) = f(\mathcal{D}\hat{\mathbf{W}}_{\mathbf{D}}^{\mathbf{T}}[\mathbf{n}]) + f(\mathcal{C}\hat{\mathbf{W}}_{\mathbf{C}}^{\mathbf{T}}[\mathbf{n}])$

Thus, choose $\hat{\mathbf{W}}_{\mathbf{D}}^{\mathbf{T}}[\mathbf{n}] = \mathbf{W}_{\mathbf{D}}^{\mathbf{T}}[\mathbf{n}]$ and $\hat{\mathbf{W}}_{\mathbf{C}}^{\mathbf{T}}[\mathbf{n}] = \mathbf{W}_{\mathbf{C}}^{\mathbf{T}}[\mathbf{n}]$ such that $h[n] = \hat{h}[n]$

② $o_D[n] < 0 < o_C[n]$ and $|o_C[n]| > |o_D[n]|$

$h[n] = f(o_D[n] + o_C[n]) = o_D[n] + o_C[n] = \mathcal{D}\mathbf{W}_{\mathbf{D}}^{\mathbf{T}}[\mathbf{n}] + \mathcal{C}\mathbf{W}_{\mathbf{C}}^{\mathbf{T}}[\mathbf{n}]$

$\hat{h}[n] = f(\hat{o}_D[n]) + f(\hat{o}_C[n]) = f(\mathcal{D}\hat{\mathbf{W}}_{\mathbf{D}}^{\mathbf{T}}[\mathbf{n}]) + f(\mathcal{C}\hat{\mathbf{W}}_{\mathbf{C}}^{\mathbf{T}}[\mathbf{n}])$

If $\hat{\mathbf{W}}_{\mathbf{D}}^{\mathbf{T}}[\mathbf{n}] = \mathbf{W}_{\mathbf{D}}^{\mathbf{T}}[\mathbf{n}] \Rightarrow f(\hat{o}_D[n]) = 0 \Rightarrow \hat{h}[n] = f(\mathcal{C}\hat{\mathbf{W}}_{\mathbf{C}}^{\mathbf{T}}[\mathbf{n}])$

Assume $\mathcal{C}\hat{\mathbf{W}}_{\mathbf{C}}^{\mathbf{T}}[\mathbf{n}] > 0 \Rightarrow f(\mathcal{C}\hat{\mathbf{W}}_{\mathbf{C}}^{\mathbf{T}}[\mathbf{n}]) = \mathcal{C}\hat{\mathbf{W}}_{\mathbf{C}}^{\mathbf{T}}[\mathbf{n}]$

We have $\mathcal{C}\hat{\mathbf{W}}_{\mathbf{C}}^{\mathbf{T}}[\mathbf{n}] = \mathcal{D}\mathbf{W}_{\mathbf{D}}^{\mathbf{T}}[\mathbf{n}] + \mathcal{C}\mathbf{W}_{\mathbf{C}}^{\mathbf{T}}[\mathbf{n}]$ which is an underdetermined system of equations

We know $\mathcal{C} \neq \bar{0}$ by design (e.g., one-hot encoded vector)

Thus, there exists a $\hat{\mathbf{W}}_{\mathbf{C}}^{\mathbf{T}}[\mathbf{n}]$ that satisfies the above equation such that $h[n] = \hat{h}[n]$

③ $o_D[n] < 0 < o_C[n]$ and $|o_C[n]| < |o_D[n]|$

$h[n] = f(o_D[n] + o_C[n]) = 0$

$\hat{h}[n] = f(\hat{o}_D[n]) + f(\hat{o}_C[n])$

Thus, choose $\hat{\mathbf{W}}_{\mathbf{D}}^{\mathbf{T}}[\mathbf{n}] = \mathbf{W}_{\mathbf{D}}^{\mathbf{T}}[\mathbf{n}]$ and $\hat{\mathbf{W}}_{\mathbf{C}}^{\mathbf{T}}[\mathbf{n}] = \bar{0}$ such that $h[n] = \hat{h}[n] = 0$

④ $o_D[n] < 0$ and $o_C[n] < 0$

$h[n] = f(o_D[n] + o_C[n]) = 0$

$\hat{h}[n] = f(\hat{o}_D[n]) + f(\hat{o}_C[n])$

Thus, choose $\hat{\mathbf{W}}_{\mathbf{D}}^{\mathbf{T}}[\mathbf{n}] = \mathbf{W}_{\mathbf{D}}^{\mathbf{T}}[\mathbf{n}]$ and $\hat{\mathbf{W}}_{\mathbf{C}}^{\mathbf{T}}[\mathbf{n}] = \mathbf{W}_{\mathbf{C}}^{\mathbf{T}}[\mathbf{n}]$ such that $h[n] = \hat{h}[n] = 0$

⑤ $o_C[n] < 0 < o_D[n]$ and $|o_D[n]| < |o_C[n]|$

$h[n] = f(o_D[n] + o_C[n]) = 0$

$\hat{h}[n] = f(\hat{o}_D[n]) + f(\hat{o}_C[n])$

Thus, choose $\hat{\mathbf{W}}_{\mathbf{D}}^{\mathbf{T}}[\mathbf{n}] = \bar{0}$ and $\hat{\mathbf{W}}_{\mathbf{C}}^{\mathbf{T}}[\mathbf{n}] = \mathbf{W}_{\mathbf{C}}^{\mathbf{T}}[\mathbf{n}]$ such that $h[n] = \hat{h}[n] = 0$

⑥ $o_C[n] < 0 < o_D[n]$ and $|o_C[n]| < |o_D[n]|$

$h[n] = f(o_D[n] + o_C[n]) = o_D[n] + o_C[n] = \mathcal{D}\mathbf{W}_{\mathbf{D}}^{\mathbf{T}}[\mathbf{n}] + \mathcal{C}\mathbf{W}_{\mathbf{C}}^{\mathbf{T}}[\mathbf{n}]$

$\hat{h}[n] = f(\hat{o}_D[n]) + f(\hat{o}_C[n]) = f(\mathcal{D}\hat{\mathbf{W}}_{\mathbf{D}}^{\mathbf{T}}[\mathbf{n}]) + f(\mathcal{C}\hat{\mathbf{W}}_{\mathbf{C}}^{\mathbf{T}}[\mathbf{n}])$

If $\hat{\mathbf{W}}_{\mathbf{C}}^{\mathbf{T}}[\mathbf{n}] = \mathbf{W}_{\mathbf{C}}^{\mathbf{T}}[\mathbf{n}] \Rightarrow f(\hat{o}_C[n]) = 0 \Rightarrow \hat{h}[n] = f(\mathcal{D}\hat{\mathbf{W}}_{\mathbf{D}}^{\mathbf{T}}[\mathbf{n}])$

Assume $\mathcal{D}\hat{\mathbf{W}}_{\mathbf{D}}^{\mathbf{T}}[\mathbf{n}] > 0 \Rightarrow f(\mathcal{D}\hat{\mathbf{W}}_{\mathbf{D}}^{\mathbf{T}}[\mathbf{n}]) = \mathcal{D}\hat{\mathbf{W}}_{\mathbf{D}}^{\mathbf{T}}[\mathbf{n}]$

We have $\mathcal{D}\hat{\mathbf{W}}_{\mathbf{D}}^{\mathbf{T}}[\mathbf{n}] = \mathcal{D}\mathbf{W}_{\mathbf{D}}^{\mathbf{T}}[\mathbf{n}] + \mathcal{C}\mathbf{W}_{\mathbf{C}}^{\mathbf{T}}[\mathbf{n}]$ which is an underdetermined system of equations

We know $\mathcal{C} \neq \bar{0}$ by design (e.g., one-hot encoded vector)

Thus, there exists a $\hat{\mathbf{W}}_{\mathbf{D}}^{\mathbf{T}}[\mathbf{n}]$ that satisfies the above equation such that $h[n] = \hat{h}[n]$

As a result, we can find a $\hat{\mathbf{W}}_{\mathbf{D}}^{\mathbf{T}}$ and a $\hat{\mathbf{W}}_{\mathbf{C}}^{\mathbf{T}}$ that satisfies $h = \hat{h}$. It is trivial to show that the same analysis holds for non-input layers, where there is a single term for $h$, similar to Equation 3. Finally, we use the following Python function to automatically calculate the weights of the GIFF layer from an input FF layer using the above six regions and corresponding equations:

```python
1   import torch
2   def calc_GIFF_weights(layerFF, layerGIFF, loader):
3       ww = layerFF.fc.weight.detach()
4       # Slice the weights. In this example D=120, C=8
5       wD = ww[:,0:120]
6       wC = ww[:,120:]
7
8       for batch_no, (x, y_ground) in enumerate(loader):
9           D, C = x.float().to(opts.device).view(1, -1), y_ground.float().to(opts.device)
10          # Input for FF
11          DC = torch.cat((D, C), dim=1)
12          # Pass data through FF and obtain FF activations
13          h1 = layerFF(DC)
14          # Obtain OD and OC
15          oD = torch.matmul(D, wD.T)
16          oC = torch.matmul(C, wC.T)
17          # Initialize weight matrices for the GIFF layer
18          whatD = torch.zeros_like(wD, dtype=torch.float)
19          whatC = torch.zeros_like(wC, dtype=torch.float)
20          # iterate over neurons
21          for i, oDn in enumerate(oD[0,:]):
22              oCn = oC[0,i]
23              # region 1
24              if torch.ge(oDn,0) and torch.ge(oCn,0): # ge: greater or equal
25                  whatD[i] = wD[i]
26                  whatC[i] = wC[i]
27              # region 2
28              elif torch.lt(oDn,0) and torch.gt(oCn,0) and torch.ge(oDn+oCn,0): # lt: less than
29                  whatD[i] = wD[i]
30                  whatC[i] = wC[i] + torch.matmul(torch.inverse(torch.matmul(C.T,C)), torch.matmul(C.T,oD))
31              # region 3
32              elif torch.lt(oDn,0) and torch.gt(oCn,0) and torch.le(oDn+oCn,0):
33                  whatD[i] = wD[i]
34                  whatC[i] = 0
35              # region 4
36              elif torch.lt(oDn,0) and torch.lt(oCn,0):
37                  whatD[i] = wD[i]
38                  whatC[i] = wC[i]
39              # region 5
40              elif torch.lt(oCn,0) and torch.gt(oDn,0) and torch.lt(oDn+oCn,0):
41                  whatD[i] = 0
42                  whatC[i] = wC[i]
43              # region 6
44              elif torch.lt(oCn,0) and torch.gt(oDn,0) and torch.ge(oDn+oCn,0):
45                  whatD[i] = wD[i] + torch.matmul(torch.inverse(torch.matmul(D.T,D)), torch.matmul(D.T,oC))
46                  whatC[i] = wC[i]
47              else:
48                  print("[ERROR] ERROR!")
49
50          # Write weights into the GIFF layer
51          layerGIFF.fc1.weight.data = whatD
52          layerGIFF.fc2.weight.data = whatC
53
54          # Pass data through GIFF and obtain GIFF activations
55          h2, _, _ = layerGIFF(D, C)
56
57          # Is h1 == h2 ??
```

Listing 1: Python function to calculate weights of a GIFF layer from an FF layer.

## A.2 PSEUDOCODE FOR FF AND GIFF INFERENCE LOOPS

**Algorithm A1:** Pseudocode for FF inference. A sample $\mathcal{D}$ goes through the network $C$ times.

```
1  def inference_FF():
       Input: 𝒟 – data sample, 𝒞 – vector of labels in the dataset, net – FF network with L layers
       Output: p – predicted label
2      for c in 𝒞:
           #Embed c in 𝒟
3          𝒟 ← embed(𝒟,c)
           #Pass the embedded data through the network and obtain
            activations
4          H ← net.forward(𝒟)
           #Calculate the goodness for each layer
5          g ← goodness(H)
           #Save the total excitation of the network for label c
6          G[c] ← ∑ g
       #The label with the highest excitation is the prediction
7      p ← argmax G
```

---

**Algorithm A2:** Pseudocode for GIFF inference. A sample $\mathcal{D}$ goes through the network *once*.

---

1 **def** `precompute():` `#Pre-compute the activations in the separated label`
    `channel`
      **Input:** $\mathcal{C}$ – vector of labels in the dataset, $net$ – GIFF network with $L$ layers
      **Output:** $H_A$ – label activations
2     **for** $c$ *in* $\mathcal{C}$**:**
         `#Pass the labels through the separated label channel and obtain`
          `label activations`
3          $H \leftarrow net.\textbf{forward\_label}(c)$
         `#Append the activations for label` $c$ `to` $H_A$
4          $H_A.\textbf{append}(H)$
5 **def** `inference_GIFF():`
      **Input:** $\mathcal{D}$ – data sample, $\mathcal{C}$ – vector of labels in the dataset, $net$ – GIFF network with $L$ layers, $H_A$ –
           pre-computed label activations
      **Output:** $p$ – predicted label
      `#Pass the data through the data channel and obtain data activations`
6     $H_D \leftarrow net.\textbf{forward\_data}(\mathcal{D})$
7     **for** $c$ *in* $\mathcal{C}$**:**
         `#Merge the data and label activations`
8          $H \leftarrow H_D \bigoplus H_A[c]$
         `#Calculate the goodness for each layer`
9          $g \leftarrow \textbf{goodness}(H)$
         `#Save the total excitation of the network for label` $c$
10         $G[c] \leftarrow \sum g$
      `#The label with the highest excitation is the prediction`
11     $p \leftarrow \text{argmax } G$

---

### A.3   GIFF-SPECIFIC PARAMETERS AND SWEEPS

We sweep the hyperparameters specific to GIFF as follows:

- **Threshold parameter $\theta$ in the loss function (Equation 2):** $\theta$ separates positive and negative samples. The optimizer pushes the goodness of the positive samples above $\theta$, and the goodness of the negative samples below it. As a result, the choice of $\theta$ has direct implications on the learned weights and model performance. We use five different $\theta$ values for each task, as listed in Table A1.
- **Label channel width:** We can introduce redundancy to the separated label channel by employing wider layers. If the latent labels do not match the size of the data activations, a downsampling layer is needed (e.g., pooling) before the merging operator. So, this redundancy can increase the model's performance at the cost of a higher parameter count. We sweep the label channel widths for each task, as shown in Table A1.
- **Layer grouping:** By default, GIFF trains each layer individually. That is, every layer has distinct label channels, goodness and loss functions, and optimizers. GIFF also enables training layers in groups to reduce parameter count and memory requirements. For example, a network with three layers can be grouped in four ways. **(1)** each layer is trained individually (the default): {1},{2},{3}; **(2)** two groups as {1,2},{3}; **(3)** two groups as {1},{2,3}; **(4)** one group {1,2,3}. The number of possible groups explodes for deep networks with many layers, so exhausting all combinations is impractical. Grouping the deeper layers together and leaving the earlier layers in smaller groups leads to lower parameter counts and memory, as reducing the label channels for deeper layers eliminates many parameters. Thus, we employ five different grouping strategies listed in Table A1.

Table A1: Table of configurations for grouping, label channel width, and $\theta$ used in our experiments.

| | ResNet-8 | MobileNetV1 | DS-CNN |
|---|---|---|---|
| $\theta$ | 4, 6, 8, 12, 16 | 4, 6, 8, 10, 12 | 0.5, 1, 4, 8, 16 |
| **Label channel width** | $1\times, 2\times, 3\times, 4\times, 5\times$ | $2\times, 3\times, 4\times, 6\times, 8\times$ | $1\times, 2\times, 3\times, 5\times, 7\times$ |
| **Grouping** | [{1},{2},{3}], [{1,2},{3}], [{1},{2,3}] | [{1},{2},....,{12},{13}], [{1,2,3},{4,5,6,7},{8,9,10,11,12,13}], [{1,2,3,4},{5,6,7,8,9},{10,11,12,13}], [{1,2,3},{4,5,6,7},{8,9,10,11},{12,13}], [{1,2},{3,4,5,6},{7,8,9,10},{11,12,13}] | [{1},{2},{3},{4}], [{1},{2,3,4}], [{1,2},{3,4}], [{1,2},{3},{4}], [{1},{2,3},{4}] |

### A.4 Memory estimation technique and its accuracy on an example

Measuring the memory consumption of a model during training in a data center is not feasible and is not representative of a tinyML use case. Instead, we estimate the memory requirements for training the BP and GIFF networks. For a given layer $l$:

$$mem(l) = Param(l) + Grad(l) + Act(l) + Err(l) \tag{8}$$

where $Param(l)$ is the tensor that stores the parameters in layer $l$, $Grad(l)$ is the tensor that stores the gradients, $Act(l)$ denotes the activations, and $Err(l)$ is the backpropagation errors in layer $l$. Thus, the peak memory consumption for BP is the sum of the memory consumption of all of its layers $L$:

$$mem_{BP}^{Tot} = \sum_{l=1}^{L} mem(l) = \sum_{l=1}^{L} Param(l) + Grad(l) + Act(l) + Err(l) \tag{9}$$

In contrast, the peak memory consumption for GIFF is lower since GIFF can reuse the same memory space for gradients, activations, and errors of different layers. Thus, GIFF's peak memory consumption is estimated by:

$$mem_{GIFF}^{Tot} = \sum_{l=1}^{L} Param(l) + \max_l\{Grad(l) + Act(l) + Err(l)\} \tag{10}$$

Next, we illustrate that our estimations are within 5% of a C implementation where memory allocations are manually implemented. We use the same MLP network used in the motivational example in Section 3.4:

BP and GIFF data channel:
$$self.fc1 = nn.Linear(784, 1000)$$
$$self.fc2 = nn.Linear(1000, 1000)$$
$$self.fc3 = nn.Linear(1000, 1000)$$

GIFF label channel:
$$self.label\_channel_1 = nn.Linear(10, 1000)$$
$$self.label\_channel_2 = nn.Linear(10, 1000)$$
$$self.label\_channel_3 = nn.Linear(10, 1000)$$

We obtained the values in Table 2 as follows:

**BP:**

$$Grad = tensor(784*1000) + tensor(1000*1000) + tensor(1000*1000) = 2.784M * 4B = 11.136 \text{ MB}$$
$$Act = tensor(784) + tensor(1000) + tensor(1000) = 2.784k * 4B = 11.14 \text{ kB}$$
$$Err = tensor(1000) + tensor(1000) + tensor(1000) = 3k * 4B = 12 \text{ kB}$$
$$Param = tensor(784*1000) + tensor(1000*1000) + tensor(1000*1000) = 2.784M * 4B = 11.136 \text{ MB}$$
$$Mem_{BP}^{Tot} = Grad + Act + Err + Param = 22.295 \text{ MB}$$

**GIFF:** From the network architecture, we know that the second and third layers have more parameters than the first. Since they are equal, the below calculation is correct for either the second or third layer:

$$Grad = tensor(1000 * 1000) + tensor(10 * 1000) = 1.01M * 4B = 4.04 \text{ MB}$$
$$Act = tensor(1000) + tensor(10) = (1k + 10) * 4B = 4.01 \text{ kB}$$
$$Err = tensor(1000) + tensor(1000) = 2k * 4B = 8 \text{ kB}$$
$$Param = tensor(784 * 1000) + 2 * tensor(1000 * 1000) + 3 * tensor(10, 1000)$$
$$= 2.814M * 4B = 11.256 \text{ MB}$$
$$Mem_{GIFF}^{Tot} = Grad + Act + Err + Param = 15.31 \text{ MB}$$

To validate these values, we used the open-source KANN[2] library to implement the same networks in C and compiled the program *without any compiler optimizations in debug mode*. The pure C implementation allows us to carefully analyze the memory utilization of the BP and GIFF models. According to our results, BP training requires 22.437 MB and GIFF training requires 16.112 MB. Thus, our memory estimation is within 1% and 5% of the values obtained by C implementations of the BP and GIFF algorithms, respectively. Estimation for GIFF training has higher deviation than BP because we ignore the effect of optimizers in our estimations. We also ignore the effect of intermediate variables for non-learnable operators (e.g., pooling, norm). In any case, our memory estimation method gives reliable and realistic estimates; thus, we employ it for our results in the current work.

---

[2]https://github.com/attractivechaos/kann

## A.5 RESNET-8 LAYER GROUPING

Figure A3 illustrates the groups for the ResNet-8 model. The ResNet-8 has an initial convolutional layer, followed by a residual block with two convolutional layers and then two residual blocks with three convolutional layers. We always place the initial convolutional layer and the first residual block in the same group. As a result, we have a maximum of three groups, which is depicted in Figure A3a). In Figure A3b), the second and third residual blocks are in the same group, leading to two groups. In Figure A3c), the initial convolutional layer and the first and second residual blocks are in the same group, again leading to two groups.

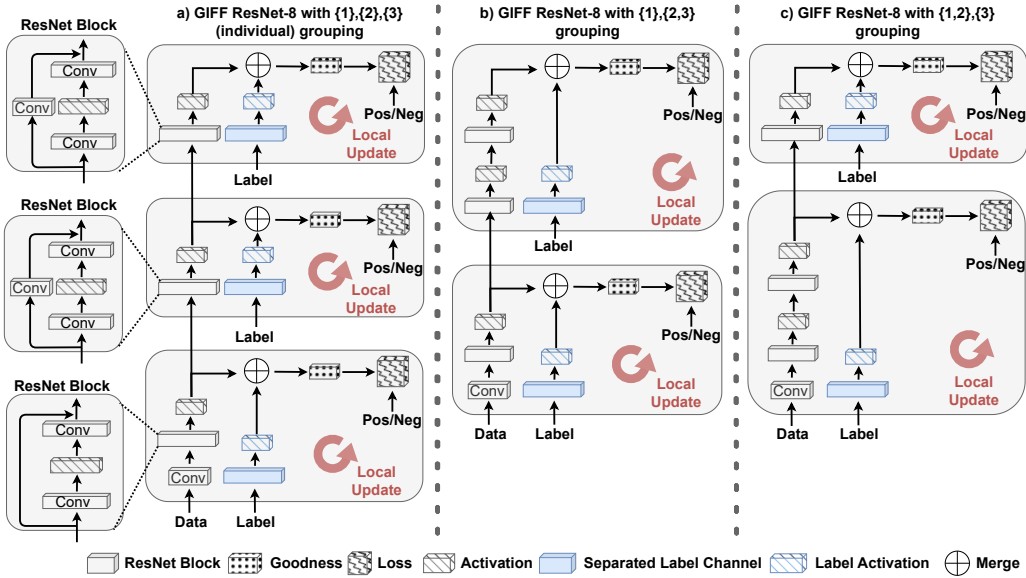

Figure A3: Three different layer grouping approaches for the ResNet-8 model: a) Individual groups, b) two groups as {1},{2,3}, and c) two groups as {1,2},{3}

For each of these groups, we can have different separated label channel widths and $\theta$s. We sweep their values to obtain numerous GIFF models that span the entire design space and thus, yield various accuracy results, as depicted in the main text.

## A.6 MobileNetV1 and ds-cnn layer grouping

Depthwise Separable (DS) Convolutional Neural Networks (DS-CNNs) consist of " blocks" containing a depthwise convolution layer followed by a point-wise convolution layer, called the depthwise separable convolution operation. In a depthwise convolution layer, channels in input share the same filter for an output channel. Pointwise convolution layer is a regular convolutional layer with the kernel size of $1 \times 1$. This DS structure significantly reduces the number of multiplication operations and enables deeper networks. It also increases the design space for layer groups for GIFF.

MobileNetV1 is a standardized DS-CNN model with particular kernel sizes for each layer. It consists of an initial convolutional layer followed by thirteen "mobile blocks". Similar to ResNet-8, the initial convolution layer and the first mobile block are placed in the first group. Therefore, the maximum number of groups is 13 for the MobileNetV1 model, as illustrated in Figure A4a). Similarly, Figure A4b) and c) illustrate two examples with three and four groups, respectively.

The DS-CNN model used for the keyword spotting benchmark has four blocks, leading to a maximum of four groups for GIFF implementation. Similar to ResNet-8 and MobileNetV1, we show three example groups for the DS-CNN model in Figure A5.

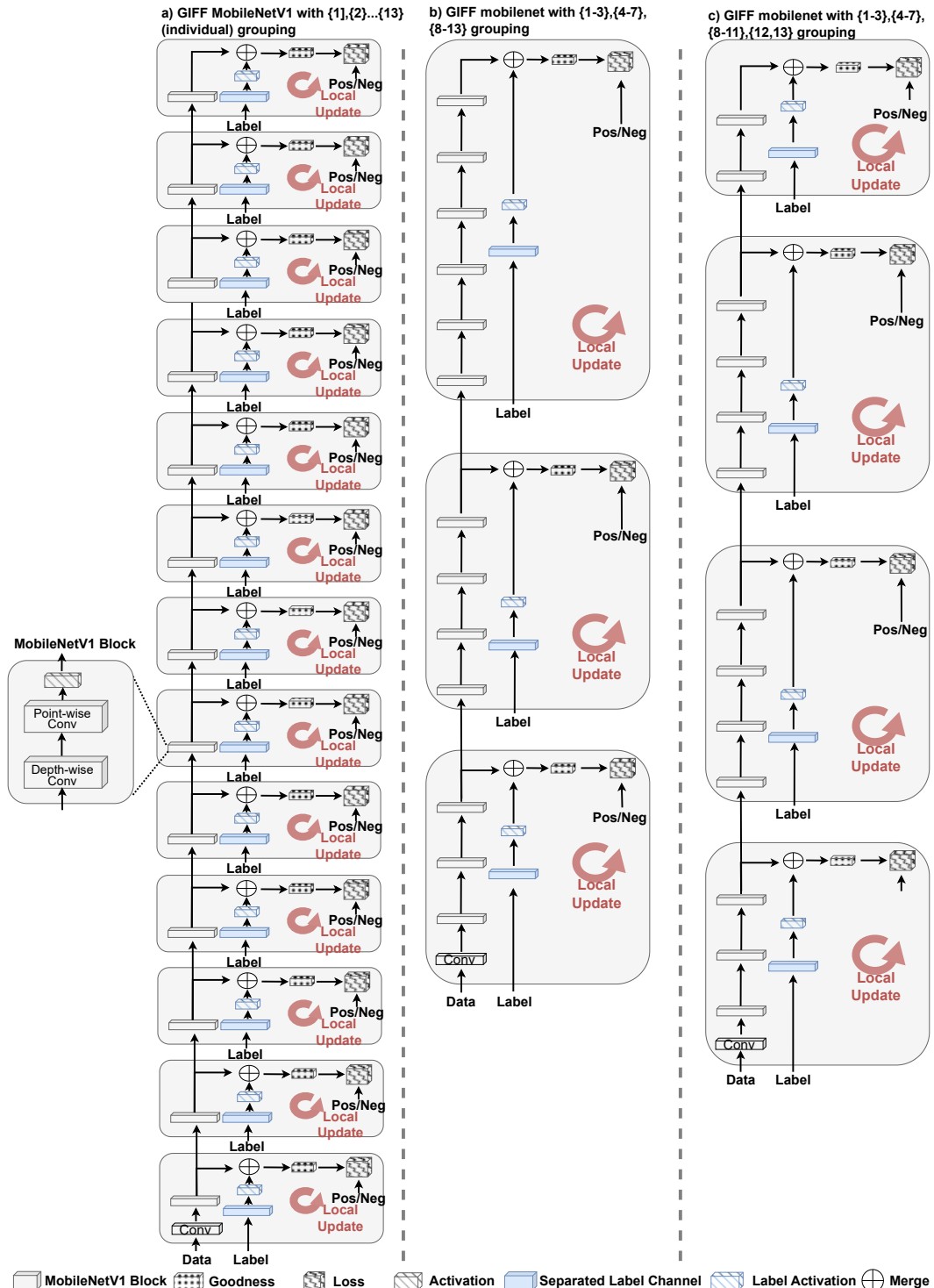

Figure A4: Three different layer grouping approaches for the MobileNetV1 model: a) Individual groups, b) three groups as {1-3},{4-7},{8-13} and c) four groups as {1-3},{4-7},{8-11},{12,13}

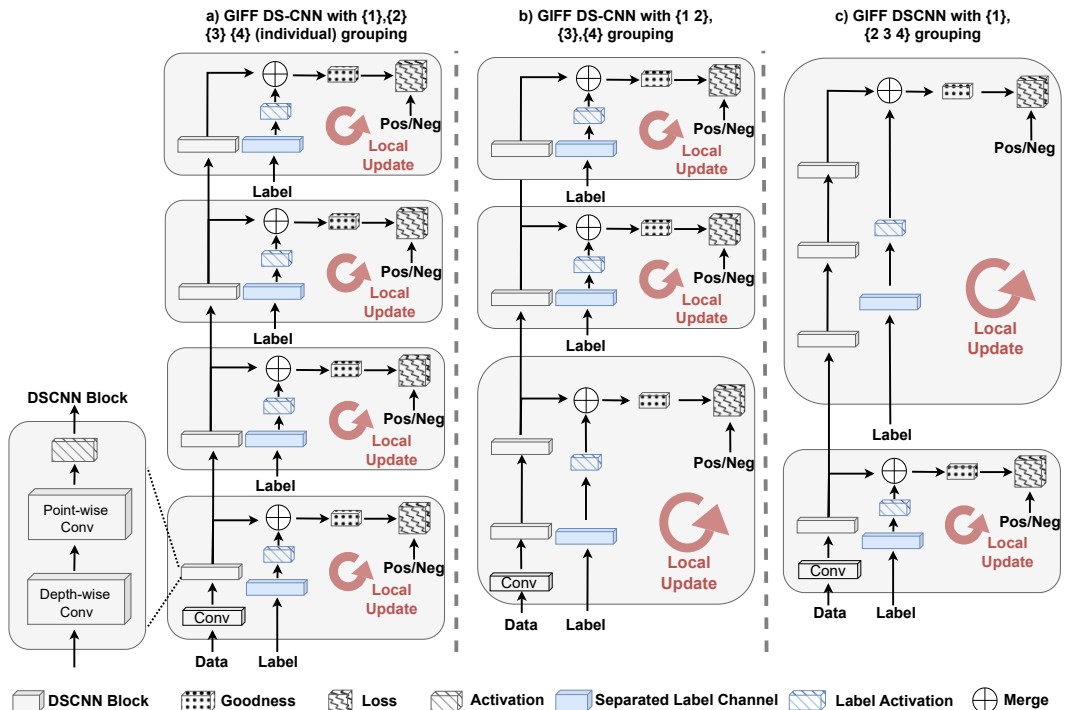

Figure A5: Three different layer grouping approaches for the DS-CNN model: a) Individual groups, b) three groups as {1,2},{3},{4} and c) two groups as {1},{2-4}

