# OpenReview forum: "GIFF: Generalized Inference Friendly Forward-Forward Algorithm"
_ICLR.cc/2024/Conference — ICLR 2024 Conference Withdrawn Submission_

### Official Review · Reviewer_V3Fd · 2023-10-26

**Soundness:** 3 good
**Presentation:** 3 good
**Contribution:** 3 good
**Rating:** 5
**Confidence:** 4

**Summary:**

The present paper proposes a variant of the Forward-Forward (FF) algorithm [Hinton, 2022] which circumvents three issues: 1/ how the label is embedded and fed to the network, 2/ the poor performance of FF with conv layers, 3/ the inefficiency of the FF inference, which requires C forward passes for each class prediction, where C denotes the number of classes. The authors solve all three by proposing a separate label forward pathway, whereby the label is no longer fed only at the input layer through a single common data pathway, but broadcasted to all layers and embedded with small local auxiliary nets. This way, activations of the data forward pathway only need to be computed once and only the forward passes through the auxiliary nets which embed the label are computed C times. The authors claim superior performance to the standard FF algorithm, smaller memory footprint than BP and FLOPs for inference based on TinyML benchmarks (image classification, person detection and keyword spotting) on convolution-based architectures.

More precisely:
- Section 3.1 and 3.2 reintroduce backprop and the FF algorithm.
- Section 3.3 presents the core GIFF architectural ingredients, namely 1/ the separation of the feedforward pathway into one for the data, another one for the labels, 2/ a "merging operation" to blend the data activation and label information which is fed into the goodness function.
- Section 3.4 presents results obtained on MNIST on MLPs, Convnets for GIFF, FF and BP in terms of number of parameters, peak memory and accuracy. It is claimed that GIFF maintains accuracy on par with BP while requiring much less memory (Table 2). Also it is shown that GIFF requires 10 times less MFLOPs than FF, as expected by construction of GIFF.
- Section 4.1 introduces the experimental setup and provides a summary of the results obtained on the CIFAR10, VWW and GSC tasks for BP and GIFF.
- Section 4.2 presents results obtained on CIFAR10 with. the ResNet-8 architecture, presenting the test accuracy vs peak memory tradeoff for various layer groupings (i.e. one auxiliary net for the label embedding might be used jointly for several layers, which reduces memory footprint of these auxiliary nets). It also shown that GIFF requires 10 times less MFLOPs than FF (as reported previously).
- Section 4.3 and 4.4 report the same kind of study on the GSC and VWW tasks.

**Strengths:**

- The paper is well written.
- The use of auxiliary nets for the separate label forward pathway undeniably solves the inference problem of the FF algorithm.
- GIFF is tested across various models and datasets, and various model setups (e.g. layer groupings).
- I like that an alternative to BP is motivated for tinyML with quantitative metrics rather than fuzzy biologically plausible arguments.

**Weaknesses:**

- While I am confident on the FLOPs reported for inference, I am much more skeptical on the memory peak evaluation (I detail in the next section why) and afraid it might skew all the presented results. I am really not sure about Eq. 9 of Appendix A.4. And in fact, it is up to 5% off the evaluation done by the C implementation on a tiny architecture used for MNIST. I am worried since based on this sole observation on the fully connected architecture used on MNIST, **"[they] employ [their memory estimation method] for [their] results in the current work"**. Therefore if the formula is incorrect, all the graphs (including the diamond graphs depicting accuracy vs memory peak utilization) are incorrect too.
- When using layer groupings, for example for MobileNet ((1-3), (4-7), (8-13) groupings), **backpropagation might be used inside blocks**. While the authors say that non-gradient based techniques can be used for FF-based optimization, they need to clarify how the learning *within* each block has been achieved.
- Please clarify why non-gradient based techniques (e.g. integer linear programming, evolutionary optimizers) can be used for FF-based optimization specifically.
- Some quantitative comparisons are unfair (I detail below which ones).
- No baseline experiments with the FF algorithm nor any FF variants are ever presented, and I think this is critically missing.

**Questions:**

- Consider the following computational pathway for the data:
```
     G_1   G_2   G_3
      ↑     ↑     ↑
x  → s_1 → s_2 → s_3
      ↑     ↑     ↑
    w_1    w_2   w_3
```

For simplicity, we omit the label pathway, non parametrized operations (e.g. normalization). For simplicity, let's assume that we consider a deep linear network such that $s_k = w_k \cdot s_{k-1}$, $s_0=x$, $g_k = G(s_k, l_k)$ where $l_k$ denotes some label embedding. Computing the GIFF gradients amounts to compute $\Delta w_k \propto \partial_{s_k} g_k \cdot s_{k-1}^\top$. Relating to your own terminology, the peak memory consumption for a layer $k$ takes into account $Param$ (to store $w_k$), $Grad$ (to store $\Delta w_k$), $Act$ (to store $s_k$) and $Err$ (to store $\partial_{s_k}g_k$). Below Eq. 9, you claim that "the peak memory consumption for GIFF is lower since **GIFF can reuse the same memory space for gradients, activations, and errors of different layers**". So this goes to say (if I understood correctly) that you can change *in-place* the memory allocated to neurons to store *either* activation, error signals or weight gradients.

My question is the following: could you please detail very clearly the computation pipeline (along with the memory units) for the simple example I proposed above to justify your claim and contrast it with backpropagation?

- Please state clearly how you optimize the parameters **inside blocks**, when grouping multiple layers together. **Do you use backprop?**.

- "Table 2 shows that GIFF enables higher accuracy than FF (98.4% vs. 97.2%) using 279× fewer parameters (2.82M vs. 10.1k) and 220× smaller peak memory (15.3 MB vs. 73.4 kB)". Here you compare GIFF **on a convnet** to FF **on a MLP**. Most of the memory savings merely come from using a convnet rather than using GIFF instead of FF. Please make a fair comparison based on the *same* architecture.

- A comment on Table 2 : how is this possible that GIFF and FF use the exact same number of parameters, given that GIFF uses one extra parametrized layer *per* layer to compute the label embeddings?

- You write multiple times in the paper that FF training on convnets is "not supported", is "prevented", "FF does not work with any layers that use weight-sharing". Although the statement is pretty assertive, the justification brought is a bit light and in fact there have been many studies investigating the FF algorithm on Convnets architectures, surely with a poor performance compared to backprop, but still with a decent accuracy (better than chance). I would really need to see convnets experiments with FF as well , **including most recent changes to the FF algorithm that could have improved its performance on convnets**, to be convinced that your approach truly solves this problem. Could you please explain why?

- Your FF implementation, as well as GIFF, underperforms the FF implementation of this repo (https://github.com/loeweX/Forward-Forward).

---

### Official Review · Reviewer_8xmW · 2023-10-30

**Soundness:** 2 fair
**Presentation:** 3 good
**Contribution:** 2 fair
**Rating:** 5
**Confidence:** 4

**Summary:**

This paper proposes a generalized inference friendly forward-forward (GIFF) algorithm to address the limitations of the Forward-Forward (FF) algorithm by introducing a label branch, which can tackle inefficient inference passes and support conventional networks. The experiments show that GIFF achieves performs similar results with BP and use up to 43% less memory.

**Strengths:**

- The paper introduces a label branch for FF, which can tackle inefficient inference passes and support conventional networks.

- The proposed GIFF achieves performs similar results with BP and use less memory.

**Weaknesses:**

Weaknesses:
- The proposed GIFF is similar to a network with multiple losses at different layers. For example, GoogLeNet [r1] used the losses of the auxiliary classifiers
to improve the accuracy of the classification. The main differences could be that GIFF only allows local (layer-wise) gradient.

[r1] Going deeper with convolutions, 2014.

- When applied to very deep networks, such as ResNet-8, the proposed method has to split it into several groups (2 groups in Figure A3). Within each group, it is unclear if the proposed method uses BP. If so, I do not think the proposed GIFF is a general FF method. Moreover, how to design networks heads (called Goodness in the paper? ) and losses in the lower layers? Is it very heavy? If we applied GIFF to each layer of a deep networks, is it contain lots of heavy heads?

**Questions:**

See [Weaknesses]

---

### Official Review · Reviewer_QKxL · 2023-11-01

**Soundness:** 1 poor
**Presentation:** 4 excellent
**Contribution:** 1 poor
**Rating:** 1
**Confidence:** 4

**Summary:**

Forward-Forward (FF) is a recently proposed alternative to backpropagation (BP), which does not require propagation of a global error signal across layers.
However, the authors identify 3 limitations of FF: it suffers from the label embedding problem, does not support convolutional layers, and is not efficient during inference for supervised learning tasks.
Thus, the authors propose Generalized Inference Friendly Forward-Forward (GIFF) to address these limitations.
Unlike FF, GIFF uses a separate channel to pass the label through the network; by separating the data and label channels, GIFF does not constrain the type of layer used.
The authors empirically evaluate GIFF on the TinyML benchmark, and show that it is more computationally efficient than FF, and more memory efficient than BP while achievign similar test accuracy.

**Strengths:**

- This paper is very well-written. It flows well, and the figures are clear and easy to follow.
- The authors propose an approach that potentially improves the memory and computational efficiency of training neural networks. This is clearly an important research direction, particularly given the increasing climate change concerns of training large models.

**Weaknesses:**

While this paper is well-written, I’m not convinced on the novelty or importance of its contributions.
The authors claim that they alleviate 3 limitations of FF, namely the label embedding problem, lack of support for convolutional layers, and inefficiency during inference. But none of these appear to be real limitations of FF.
- As far as I understand, the first 2 limitations both stem from the same problem, i.e. that for supervised learning, the original FF paper embeds the label as part of the input data.
However, that was an ad-hoc solution Hinton made for the purpose of demonstrating how conceptually FF can be applied to supervised learning problems, rather than an actual design choice of FF.
Since his setting was MNIST, and the border pixels of MNIST samples are not used to store useful information, he uses them to store the label, rather than code up a separate pipeline.
But there are no algorithmic limitations that prevent him from using a separate channel to store the labels; embedding them to the input was just a hacky solution for simplicity and convenience.
Thus, I don’t agree that FF *requires* label embedding, or that label embedding is an issue that prevents it from being applied to other types of layers.
To support my point— the authors claim that “GIFF can train ResNet and MobileNet models, which is not possible with FF,” supposedly because of the label embedding issue.
But in the FF paper, Hinton states “There is clearly no problem adding skip-layer connections, but the simplest architecture is the easiest to understand and the aim of this paper is understanding, not performance on benchmarks.”
- For the 3rd limitation, the authors’ claim that FF is inefficient during inference relies on the assumption that FF needs a separate inference pass for each class.
But the FF paper provides an alternative approach, which trades accuracy for efficiency: by initializing the input with a neutral label, it’s possible to get the prediction with a single inference pass, just like GIFF.
If the authors want to claim computational efficiency, I think it’s important to compare GIFF’s performance to the efficient version of FF, rather than the inefficient version.

**Questions:**

- I’m happy to be proven wrong here, but it appears to me that the main reason you claim FF does not support convolutional layers is that it embeds the label in its input. Are there any other challenges that you had to overcome to implement convolutional layers that may be significant to you contributions, after moving the label to a separate channel?

- Like FF, PEPITA (Dellaferrera & Kreiman, 2022) is another forward-only learning algorithm that utilizes 2 forward passes instead of a forward and backward pass.
As far as I’m aware, like GIFF, PEPITA does not have the label embedding problem, supports convolutional layers, and does not require multiple inference passes for classification.
Given the similarities in the advantages these approaches offer, I’d like to see the authors compare the two approaches in terms of design, performance, and limitations.
In other words, could you provide an argument of why we need GIFF given PEPITA exists?


References:

*Dellaferrera, G., & Kreiman, G. (2022, June). Error-driven input modulation: solving the credit assignment problem without a backward pass. In International Conference on Machine Learning (pp. 4937-4955). PMLR.*

---

> ### Comment · Reviewer_QKxL · 2023-12-04
>
> I am rather concerned with the lack of response from the authors, as I believe there are some serious issues with the content of the paper that may mislead future researchers if it is published at ICLR.
>
> My primary concern is that the limitations of FF that this paper purports to address are not actual limitations of FF, but stem from the authors' misunderstandings of FF. As this is a strong claim, I have re-read the paper to check if I have overlooked anything, but my position remains unchanged.
>
> In summary, my position is that:
> 1. FF is not constrained by the "label embedding problem"– Hinton embedded the labels in his supervised learning experiments out of convenience, not necessity.
> 2. Label embedding is not the reason that FF does not support convolutional layers. FF does not support convolutional layers because it is designed to run on neuromorphic hardware, which, unlike von Neumann architectures, keeps memory local to distributed compute units (much like the brain). This makes it prohibitively expensive to share weights for low-energy use.
> 3. Contrary to the authors' claim, FF can support ResNets. In fact, Hinton explicitly states that "there is clearly no problem adding skip-layer connections."
> 4. The authors claim to achieve improved efficiency over FF, but compares GIFF against the inefficient version of FF, not the efficient version.
>
> As these issues essentially invalidate the central purpose of this paper, I am decreasing my score to a strong reject pending further discussion. If the AC or other reviewers disagree with my evaluation, I'd much appreciate that they weigh in on this.

---

### Official Review · Reviewer_8ETu · 2023-11-01

**Soundness:** 3 good
**Presentation:** 3 good
**Contribution:** 3 good
**Rating:** 5
**Confidence:** 2

**Summary:**

This paper proposes GIFF to improve the existing FF algorithm that enables local optimization and continuous training on edge devices. It solves the update for convolutional layers and dependence on label embedding. However, the accuracy achieved is still obviously lower than the BP methods.

**Strengths:**

- Proposes new methods to achieve optimization process without dependence on backward propagation.
- Conduct experiments on three different tasks.
- Consider the practical inference speed and complexity.

**Weaknesses:**

- only conduct experiments on toy datasets.
- the target accuracy value is lower than the value SOTA methods achieve.

**Questions:**

- How about the method's effect on regression and generative methods?